# Using Energy Particle Detection Technology on the Tiangong's Space Station's Wentian Laboratory Cabin Module

Guohong Shen [1,2,]*, Shenyi Zhang [1,2], Xianguo Zhang [1,2], Huanxin Zhang [1,2], Bin Yuan [1,2], Donghui Hou [1,2], Chunqin Wang [1,2], Zida Quan [1,2], Zhe Yang [3] and Yueqiang Sun [1,2]

1   National Space Science Center, Chinese Academy of Sciences, Beijing 100190, China
2   Beijing Key Laboratory of Space Environment Exploration, Beijing 100190, China
3   University of Chinese Academy of Sciences, Beijing 100190, China
*   Correspondence: shgh@nssc.ac.cn; Tel.: +86-10-62560947

**Abstract:** To conduct real-time monitoring of the particle radiation environment in the orbit of the Tiangong space station, the installation of an energy particle detector operating on the outside of Wentian laboratory cabin module is proposed. Monitoring the energy, flux, and direction of high-energy protons, electrons, heavy ions, and neutrons in orbital space, as well as the LET spectrum and radiation dose rate generated by them, provides an important basis for studying the mechanism of the space environment that causes harm to space stations and astronauts. It also provides the necessary space environment parameters for the scientific experiment instruments on the space station. During its ground development process, the detector was verified by various calibration methods such as standard radioactive sources, equivalent signal generators, and particle accelerators. The results show that the detector can realize discrimination of particle ingredients (electrons, protons, heavy ions, and neutrons). Meanwhile, the measurement indexes can also realize target requirements, namely, from lower limit of 20 keV for medium-energy electrons and protons to heavy ion GeV, 0.025 eV~100 MeV for neutrons, and 0.233~17,475 keV/μm for the LET spectrum and 0.1~1000 mGy/day for the dose rate produced. The measurement precisions of all indexes are better than approximately 16%.

**Keywords:** Tiangong space station; Wentian laboratory cabin module; energy particle detector; neutron detection; heavy ion detection; LET spectrum; radiation dose





## 1. Introduction

China's space station, Tiangong, is a long-term space infrastructure of on-orbit operation and is an important platform for various space science experiments [1]. The main task of space environment monitoring is to monitor the space energy particle radiation environment in the orbit of the space station, which is one of the important tasks that ensures the safety of manned space projects. Meanwhile, long-term, on-orbit detection of space environment elements can provide an important basis for studying the mechanism of the space environment that causes harm to the space station and astronauts [2–4]. Additionally, this can also provide necessary space environment parameters for scientific experiment instruments on the space station [5].

The energy particle detector (EPD) on the Wentian laboratory cabin module of the space station was self-developed by the National Space Science Center of Chinese Academy of Sciences. It is the space particle radiation detector with the most complete particle types and the widest energy spectrum range in China to date. It is loaded with a medium energy electron detection unit (20–400 keV, 9 detection directions), a medium-energy proton detection unit (20 keV–10 MeV, 9 detection directions), a comprehensive detection unit (electrons: 0.4–10 MeV; protons: 8–300 MeV; heavy ions: 8–400 MeV/nucleon; 5 detection directions; dose rate: 0.1~1000 mGy/day; LET spectrum: 0.233~17,475 keV/μm), and a neutron detection unit (0.025 eV~100 MeV). The main goal of the EPD is to monitor the

energy, flux, and direction of high-energy protons, electrons, heavy ions, and neutrons in the orbit of the space station, as well as the LET spectrum and radiation dose rate generated by them, thereby providing key reference data for the safety of the space station, astronauts going out of the cabin, and space science experiments. Figure 1 shows a photo of the EPD on the Tiangong space station's Wentian laboratory cabin module.

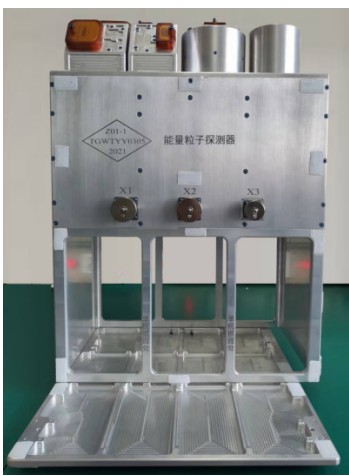

**Figure 1.** Photo of energy particle detector on the Tiangong space station's Wentian laboratory cabin module.

## 2. Scientific Objectives

### 2.1. Radiation Environment of Orbital Particles

The Tiangong space station operates in a near-circular orbit at an altitude of 340–450 km and an inclination of 41–43°. The radiation in this orbit mainly comes from radiation belts, galactic cosmic rays, and occasional solar particle events [6,7]. The orbital particle radiation of the space station has a complex composition in quiet time, including protons, neutrons, heavy ions, electrons, and X-rays. Different particles have different interaction mechanisms with matter, and heavy ions and neutrons cause different levels of damage to the space station and astronauts. The particle energy range is extremely wide, from tens of keV to more than tens of GeV. High-energy particles have strong penetrating powers, are difficult to shield, and produce a large number of secondary particles after interacting with matter. In addition, orbital particle radiation is affected by many factors. For example, in space, it is affected by the geomagnetic field, and the particle radiation varies with longitude, latitude, and altitude. In time, it is affected by solar activities. Some particle radiation fluxes experience long-term changes in year solar cycle. During solar flares or geomagnetic disturbances, particle radiation also changes drastically in a short period of time. According to the results of DOSTEL located on top of the head of MATROSHKA outside the ISS, for SAA and GCR contribution, dose rate is 243 uGy/d and 267 uGy/d, and dose equivalent rate is 437 uSv/d and 828 uSv/d, respectively [8].

#### 2.1.1. Radiation Belts

The region around the Earth where energetic charged particles are bound is called the radiation belt, which includes the outer radiation belt and the inner radiation belt [9–11]. Usually, the Earth's radiation belts are all above the orbit of the space station. However, due to the declination angle between the Earth's rotational axis and the geomagnetic axis, the height of the inner radiation belt in the South Atlantic region is reduced to approximately 200 km, which is the South Atlantic Anomaly (SAA). The space station is exposed to radiation from radiation-belt protons and electrons as it passes through the SAA [12].

The particle radiation environment in the manned spaceflight orbit is mainly in the SAA of the Earth's inner radiation belt [13]. Figure 2 shows an omnidirectional flux distribution diagram of the electron energy greater than 50 keV and the proton energy

greater than 1 MeV calculated according to the AE8 and AP8 models. The maximum count of electrons with energy greater than 50 keV is $3.9 \times 10^6$, and the maximum count of protons with energy greater than 1 MeV is $2.9 \times 10^4$. These models only calculate the omnidirectional integral flux, which cannot be given as differential directional intensity and does not include the change in particle intensity when the space environment is disturbed. Upon disturbance, the intensity of the particles can vary by orders of magnitude, with durations ranging from hours to days.

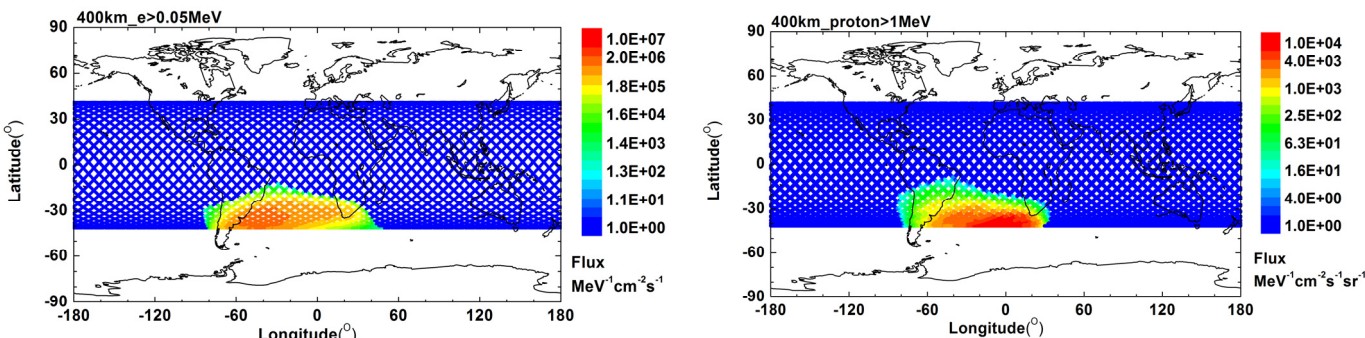

**Figure 2.** Distribution of electrons and protons in manned spaceflight orbit obtained based on AE8 and AP8 models, respectively.

### 2.1.2. Galactic Cosmic Rays

Galactic cosmic rays refer to high-energy charged particles produced outside the solar system. During the long flight, the galactic cosmic rays are deflected by the galactic magnetic field and the interplanetary magnetic field and the solar wind, and the direction of the detected cosmic rays is very different to the direction from which they originally came. The source and acceleration mechanism of the galactic cosmic rays are not fully understood [14,15].

The composition of galactic cosmic rays consists of electrons (approx. 2%) and atomic nuclei of all elements in the periodic table (approx. 98%). The energy range of galactic cosmic rays is very wide, from tens of MeV to as high as $10^{14}$ MeV [16].

Galactic cosmic rays are the main radiation exposure received by astronauts in the space station. Approximately 70–80% of the dose equivalent (a physical quantity related to radiation hazards) received by astronauts comes from galactic cosmic rays and their secondary particles [4,17]. The flux of galactic cosmic rays is low, while its energy is high, as well as being difficult to shield against. Moreover, it belongs to high LET radiation, which is seriously harmful to the human body, especially the high-charge and high-energy particles.

### 2.1.3. Solar Energetic Particle Events

Flares or coronal mass ejections occur when the Sun is violently active, accompanied by a large ejection of high-energy charged particles. High-energy charged particles include electrons, protons, and heavy ions. The occurrence of solar energetic particle events is random, and the frequency of occurrence is higher in periods of high solar activity. The composition, intensity, and energy spectrum of emitted particles are related to the location of the solar active region, magnetic field structure, and atmospheric composition. In the process of reaching the Earth, it is also affected by the Sun's magnetic field and the Earth's magnetic field [18–20].

When solar activity occurs, it causes a solar energetic particle event (SEP). An SEP may directly reach the manned spaceflight orbit and cause damage to spacecraft and personnel. Magnetic storms or other space environment disturbances caused by solar activity may affect particle radiation environment in manned spaceflight orbit. Therefore, it is necessary to forecast and monitor these physical phenomena in a timely manner, so that the spacecraft can respond accordingly in advance; for example, in large solar particle events, sensitive

instruments can be properly shut down, allowing astronauts to enter the shielded area early, and so on.

### 2.2. Particle Radiation Effect

The radiation environment of space particles causes various kinds of damage to the safe operation of spacecraft, leading to abnormal functioning of equipment inside and outside the module, and even the failure of the entire space station. It can also endanger the life and health of astronauts and their various extravehicular activities and have a major impact on space science experiments. When a spacecraft is in the space particle radiation environment, the typical effects of particle radiation received include single event effect, radiation dose effect, and charging and discharging effect on the surface or interior of the spacecraft [21,22].

A single event effect is mainly caused by high-energy protons and heavy ions. When the spacecraft is in orbit, they will change the state of the microelectronic devices of various electronic equipment on the spacecraft, which causes abnormalities or malfunctions of the equipment, resulting in single event effects [23,24].

The radiation dose effect is mainly caused by high-energy protons, electrons, and heavy ions, which can produce both ionization and displacement. When the radiation dose received by spacecraft materials or devices reaches a certain value, it causes spacecraft material denaturation and component failure [25].

The charging and discharging effect on the surface or interior of the spacecraft is mainly caused by high-energy electrons. The interaction between electrons and the spacecraft can cause the spacecraft to charge. When the spacecraft is charged to a certain extent, the electromagnetic radiation generated will interfere with the normal operation of various electrical appliances on the spacecraft, and even cause the spacecraft to fail [26].

In addition, space radiation is known or hypothesized to cause deleterious effects to the central nervous and cardiovascular systems, as well as induction of late cancers [27].

Therefore, to ensure the safety of the space station and astronauts, it is necessary to measure the high-energy particle radiation environment in the orbit of the manned space station, and to monitor and report the disturbance of the space particle environment.

### 2.3. Main Scientific Objectives

To summarize, the main task of the EPD on the Wentian laboratory cabin module is to monitor the energy, flux, and direction of high-energy protons, electrons, heavy ions, and neutrons in the orbit of the space station, as well as the LET spectrum and radiation dose rate produced by them. Its main scientific objectives are included in the following four aspects.

- Firstly, to achieve real-time monitoring and data accumulation of the orbital particle radiation energy spectrum, high-precision particle composition, fine LET spectrum, and high-sensitivity dose rate of the manned space station.

According to the characteristics of the resources, orbit, and attitude of the space station's experimental module, the EPD of the Wentian laboratory cabin module will perform comprehensive particle radiation detection. The measurement types include protons, electrons, heavy ions, and neutrons. Measuring energy ranges from medium energy to high energy. The measured physical quantities include energy spectrum, flux, directional characteristics, LET spectrum, and dose rate. As a result, a comprehensive measurement of the energy particle radiation environment in the orbit of the space station is carried out. The status of the space environment of the space station is obtained in real time, thereby assessing the environmental risk level and conducting alarm services in real time and dynamically, helping formulate and adjust the task plan and predetermined plan of the space station in time, and reducing the risk of hazards posed by the space environment.

- Secondly, to provide the measured data of the orbital particle radiation environment of the space station during disaster events such as solar proton events and high-energy

electron storms, and to provide support for the analysis and evaluation of the particle radiation hazards encountered by the space station and astronauts.

Due to the long-term operation characteristics of the space station, the cumulative impact of particle radiation, orbital atmosphere, and other environmental elements cannot be ignored. The monitoring and reporting of space environment disturbance events can avoid the impact of the space environment on important activities such as astronauts' extravehicular activities, space station flight control management, rendezvous and docking, and reentry and return. Space environment monitoring can provide (i) a basis for evaluating the space environment effects of astronauts and space stations and analyzing anomalies, (ii) detailed high-energy charged particle energy spectrum information for astronauts' extravehicular activities and the safe operation of the space station, (iii) environmental background data for the analysis of various abnormalities or failures of spacecraft, and (iv) direct evidence for the analysis of transient and gradual environmental effects caused by the space environment [28].

- Thirdly, to provide monitoring data of space environment elements for space station operation, as well as application and test (experimental) tasks, which can be used for basic research on the space environment.

One of the important missions of the space station is to provide a platform for new material experiments and life experiments. It is a testing ground for humans to develop new materials, be free from the Earth's environmental adaptability, and transform new species. Particle radiation is one of the most important environmental elements of the difference between outer space and the ground. Accurate and quantitative particle detection will provide key research parameters for new materials and new progress in life experiments [29,30].

- Finally, to provide measured data to further improve the special radiation environment model of the orbit of the space station, enhance radiation risk assessment capabilities, and support radiation risk management abilities during the long-term operation of the space station.

The unique orbital characteristics, abundant resources, and long-term on-orbit operation of the space station provide a rare opportunity for systematic, comprehensive, and high-precision detection and research on the space environment and its effects. Conducting the monitoring of the main environmental parameters of the manned spaceflight orbit is of great significance for accumulating environmental data, discovering the orbital environmental situations, and establishing the orbit-related environmental model [31,32].

## 3. Main Technical Indicators

The main technical indicators of the EPD include the information shown in Table 1.

**Table 1.** Main technical indicators of energy particle detector on Wentian laboratory cabin module.

| Item | Energy Range | Detection Direction |
|---|---|---|
| Protons | 20 keV~300 MeV | Nine for medium energy and five for high energy |
| Electrons | 20 keV~10 MeV | Nine for medium energy and five for high energy |
| Heavy ions | 8 MeV/nucleon~400 MeV/nucleon | Five |
| Neutrons | 0.025 eV~100 MeV | One |
| LET spectrum | 0.233~17,475 keV/μm, >17,475 keV/μm | One |
| Dose rate | 0.1~1000 mGy/day | One |

In this table, the energy range represents the energy of incident particles that can be measured by the instrument. The detection direction refers to the ability of the EPD to measure particles from different directions in space.

## 4. System Design

To date, the EPD of the Wentian laboratory cabin module has been the space particle radiation detector with the most complete particle types and the widest energy spectrum range in China. It is composed of a medium-energy electron detection unit (MEEDU), a medium-energy proton detection unit (MEPDU), a comprehensive detection unit (CDU), a neutron detection unit (NDU), and a shared data management unit (SDMU). The four detection units adopt different detection principles to realize the measurement of many particle types and wide energy spectrum coverage. The system composition of the EPD is shown in Figure 3.

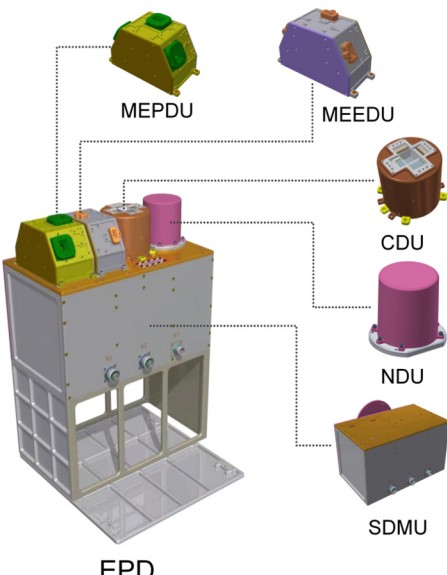

**Figure 3.** System composition of EPD.

### 4.1. Comprehensive Measurement Unit

The CDU achieves the measurement of space high-energy protons, high-energy electrons, heavy ions, neutrons, radiation LET spectrum, and dose rate. The detection unit is a hemispherical structure. The probe design adopts a multi-directional telescope system, mainly including silicon semiconductor detectors (A and B, ΔE measurement system), cesium iodide scintillator (CsI, E total energy measurement system), and plastic scintillator (anti-coincidence detection system). Among them, the ΔE semiconductor measurement system is divided into five directions. The middle direction is coaxial with the E total energy measurement system and the anti-coincidence detection system. The other four directions form an angle of 50° with the central axis and are evenly distributed on the four sides of the middle direction system. A schematic diagram of the internal structure of the probe in the CDU is shown in Figure 4.

A and B are ΔE detectors. They are used to measure the energy loss value of particles and form a fixed detection field of view. Detector A has nine sensitive areas (3 × 3 arrays) with the dimension of every array is 12 mm × 12 mm, and detector B is circular with a diameter of 12 mm. Among them, the middle area and four sensitive areas spaced apart from each other form an independent measurement field of view with the B detector. The thickness of the detector A and B is 150 μm and 300 μm, respectively.

The C detector is a CsI scintillator detector, which can stop charged particles with higher energy. It is used to measure the total energy of incident charged particles [33,34]. The shape of the CsI scintillator detector is the combination of the upper quadrangular pyramid and the lower square frustum. The output light signal is collected by the back-end light-collecting device, which is achieved by setting the light-collecting device silicon photomultiplier tube SiPM on its bottom surface.

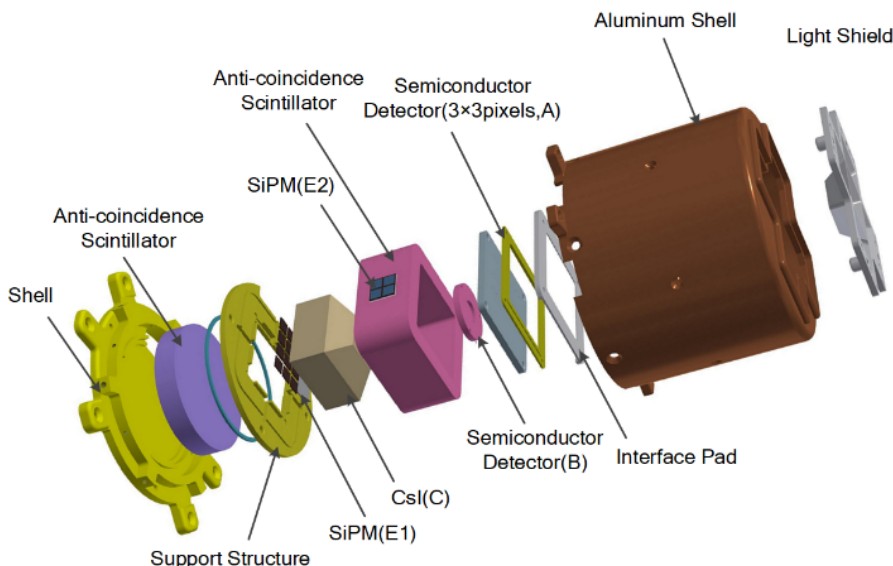

**Figure 4.** Schematic diagram of comprehensive probe structure.

As shown in Figure 5, the probes of the CDU essentially comprise two ΔE detectors (A and B), a total energy scintillator detector (C), and two anti-coincidence detectors (E1 and E2), a total of five main detectors, which record the energy loss signal of the incident particles in the sensor.

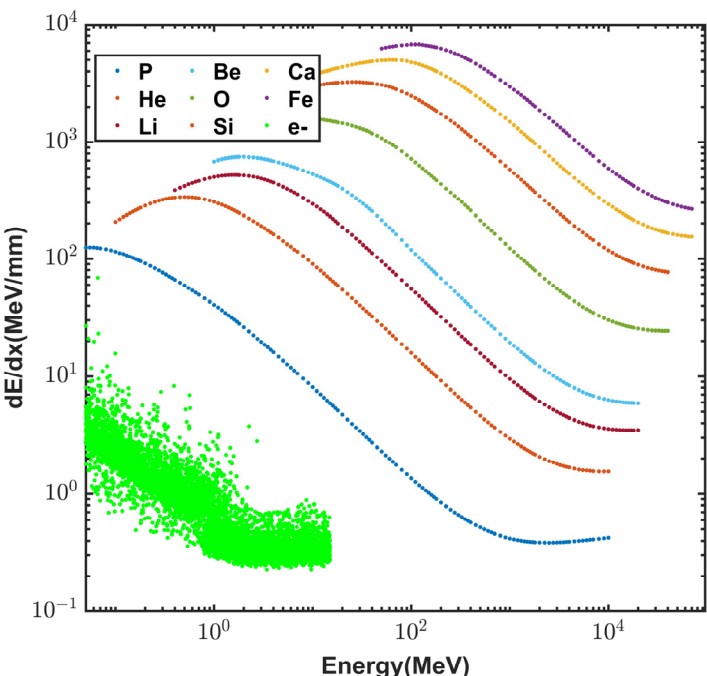

**Figure 5.** Measurement principle of ΔE•E method.

E1 and E2 are anti-coincidence detectors, which are used to identify whether the incident particle is a penetrating particle or a stopped particle. The anti-coincidence detectors are both plastic scintillators. The detectors have a certain response to charged particles, neutrons, and gamma. They can be processed into any desired shape as required. As a result, they are very suitable for making anti-coincidence detectors.

By combining the above four types of detectors, the comprehensive detection unit can achieve the measurement of high-energy particles in five directions. The measurement principle is as follows.

After the particle enters the sensor, it first loses energy in the ΔE measurement system, obtaining energy loss ΔE. After passing through ΔE, it is inserted into the total energy measurement system to obtain the total energy E. To improve the measurement precision, the ΔE measurement system uses a semiconductor silicon detector, and the total energy detector uses a CsI scintillator detector [35]. CsI has strong stopping ability and can stop and measure heavy ions with higher energy. It is currently the most used detector in the measurement of high-energy heavy ions. In terms of technology, the scintillator detector can be made very thick, and its ability to stop particles is also strong. Hence, it can measure particles with a higher energy range. To reduce the interference of obliquely incident particles and improve measurement precision, an anti-coincidence detection system is set around the detector to exclude obliquely incident particles and improve measurement precision. Moreover, there is interference from high-energy gamma (especially > 20 MeV) and high-energy protons (>tens of MeV), which can be reduced through the physical logic design of the CDU.

The basic principle of the ΔE·E method is a classical one based on the theoretical formula of particle–matter interaction. When a particle with energy E passes through a sensor with a thickness of Δx, the energy loss is ΔE, which can be expressed by the following equation [35].

$$\frac{\Delta E}{\Delta x} = k \frac{MZ^2}{E} f(v, I) \tag{1}$$

In the above equation, the parameter v and I represent the velocity of the incident particle and the average excitation energy of the material atom, respectively. By transforming it, the following formula can be obtained.

$$\Delta E \bullet E = k^{'} M Z^2 \tag{2}$$

For a specific type of particle, ΔE·E is a constant related to the atomic number of the particle. This is also the basic idea of the ΔE•E method for discrimination of heavy ions, as shown in Figure 5.

The CDU uses two silicon semiconductor detectors of the front-end ΔE system to measure the LET spectrum [36]. The LET value is the energy loss of space particles on a unit path. High-LET particles in space are considered a significant factor contributing to single-event upsets or latch-ups in electronic device. The ΔE system of CDU can be used to measure the LET value of the particle. The first piece of the position-sensitive sensor is used to measure the energy loss ΔE, and the second piece of the sensor is used as a coincidence detector. The particle is inserted into the silicon detector, and the energy ΔE (keV) is lost in the silicon detector. Knowing the thickness (d, μm) of the silicon detector, the LET value of the particle in the silicon material is as follows:

$$LET = \Delta E / d \tag{3}$$

that is, the energy loss value of the particle per unit length in the silicon material.

Meanwhile, the CDU uses two silicon semiconductor detectors at the front end to measure radiation dose. The definition of radiation dose is as follows:

$$D = \Delta E / \Delta M \tag{4}$$

that is, the total energy deposition of particles in unit mass. The radiation dose rate is as follows:

$$H = D / t \tag{5}$$

that is, the cumulative dose per unit time. The design and measurement of radiation dose rate probes start entirely from definition. Since the key components in the spacecraft

are all silicon materials, silicon materials are used as the target material for radiation dose rate measurement. Particles produce energy loss in the silicon detection material. The energy loss value ΔE of the particle is electronically recorded. The cumulative total energy loss ΣΔE per unit time is divided by the mass M of the silicon detection material to obtain the radiation dose rate. The radiation dose rate probe comprises two large-area silicon semiconductor sensors, the first for dose measurement and the second as an anti-coincidence detector.

### 4.2. Neutron Detection Unit

The EPD adopts the new material CLYC ($Cs_2LiYCl_6$: Ce) as the neutron measurement sensor for the first time in space detection. It uses the nuclear reaction between neutrons and CLYC to produce secondary particle characteristics, and inverts and measures the energy and flux of neutrons. Meanwhile, according to the particularity of neutron secondary particle waveform, the particle signal discrimination (PSD) is applied to achieve high-precision discrimination of neutron and gamma, thus effectively solving the technical problem of low neutron measurement efficiency in space particle radiation measurement [37–39].

The NDU adopts a new scintillator CLYC detector, whose chemical composition is $Cs_2LiYCl_6$: Ce. It is a new type of scintillator developed in recent years with good sensitivity to neutrons and γ rays as well as particle discrimination ability. For low-energy neutrons, they mainly react with 6Li and release 4.78 MeV energy at the same time.

$$n + {}^6Li \rightarrow \alpha + H^3 \tag{6}$$

When the neutron energy is large, the reaction cross-section with $Li^6$ is very small. Hence, fast neutrons mainly react with ${}^{35}Cl$ and release 0.615 MeV energy at the same time.

$$n + {}^{35}Cl \rightarrow {}^{35}S + P \tag{7}$$

In contrast, when the neutron energy is higher, $S^{35}$ can be in different excited energy states. $S^{35}$ in different energy states would de-excite gamma rays with different energies and allow them to be detected by CLYC, making the deposited energy continuous.

The neutron detection unit is mainly composed of CLYC crystal, anti-coincidence scintillator, photoelectric conversion multiplier device, and pre-amplification circuit, and packaged in an aluminum shell. The crystal and circuit are fixed by screws and adhesive tape. Its specific structure is shown in Figure 6.

The geometric factor of the neutron detector is approximately 40 $cm^2$sr, and the count rate is approximately 1 to 20 per second.

The most commonly used photoelectric conversion multiplier device in detectors is PMT. However, its characteristics of requiring high voltage, fragility, and large size make PMT unsuitable for requirements of low power consumption, shock resistance, and small space in the space environment. Silicon PIN photodiode and silicon photomultiplier tube SiPM can meet the requirements of the space environment. However, silicon photodiodes do not have a multiplication process, and the signal-to-noise ratio is low. In contrast, the SiPM has a very small volume. The gain can reach $10^6$ magnitudes. Its working voltage is less than 30 V. In the design, the I/V conversion circuit is used to process the SiPM output signal. The output is sent to the data-processing circuit for high-speed sampling with a frequency of 500 MHz. The anti-coincidence scintillator adopts plastic scintillator material, which can be used to eliminate the impact of protons and heavy ions on the detector in the space environment. For anti-coincidence, it is only necessary to determine whether there are charged particles entering, and there is no need to further analyze the signal. Therefore, the signal after SiPM photoelectric conversion only needs to use a comparator for threshold identification, and no amplitude sampling is required.

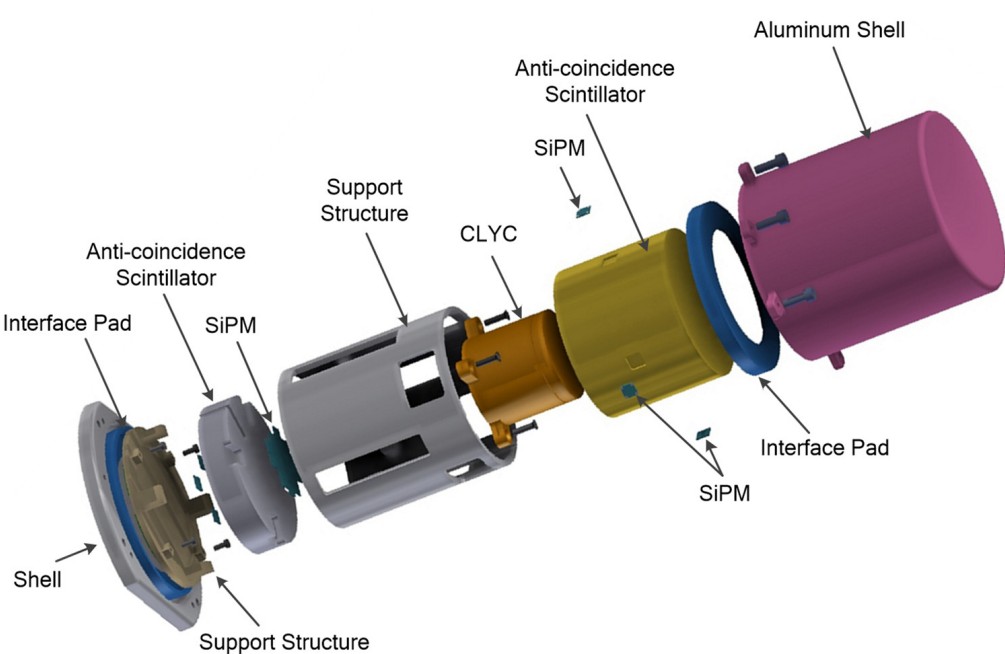

**Figure 6.** Internal structure of the neutron detection unit.

### 4.3. Medium-Energy Proton Detection Unit

The concept of the MEPDU is based on the pinhole imaging method [40,41]. Nine groups of silicon semiconductor detectors are arranged behind the precisely designed pinholes to distinguish the incident directions of medium-energy protons in nine directions with a total of 180°.

The MEPDU includes three probes. Each probe contains three sets of silicon semiconductor sensor arrays. Probes 1 and 3 are arranged in a fan shape, and probe 2 is vertical to the fan. The three sets of sensor arrays for each probe share an alignment system [42], where probes 1 and 3 detect the direction of sector 3, and probe 2 detects the direction perpendicular to sector 3. The detection field of view of each probe is 60°. The medium-energy proton measurement in nine directions is achieved, as is shown in Figure 7.

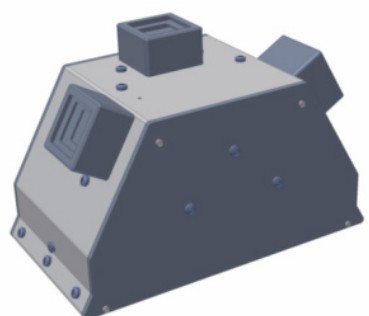

**Figure 7.** Structure of medium-energy proton detection unit.

Each probe of the MEPDU includes a front-end collimator and machine shell, a deflection magnet, a silicon semiconductor measurement system, an anti-scattering device, and a front-end data-processing circuit. Figure 8 shows a schematic diagram of the medium-energy proton structure.

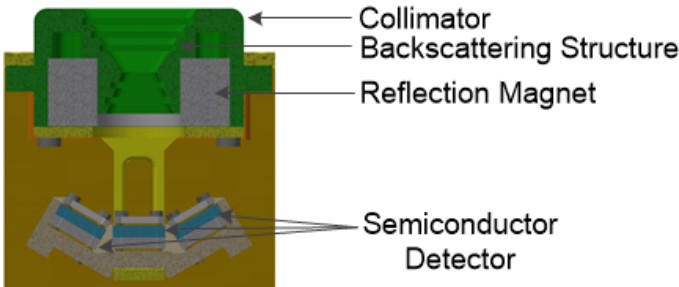

**Figure 8.** Probe section of the MEPDU.

Each direction of the medium-energy proton probe is composed of two anti-coincidence stacked semiconductor detectors, which can not only achieve the accurate measurement of the medium-energy proton energy spectrum but also shield the interference of other high-energy particles. A deflection magnet is used in the medium-energy proton probe to deflect the incident medium-energy electrons. As a result, the incident medium-energy electrons cannot enter the sensor, thereby improving the measurement precision of the medium-energy protons. The magnetic field strength is selected as 3000 Gs, and the contamination of electrons below 1 MeV can be reduced to less than 1% [43].

The main function of the MEPDU is to process and collect the particle signals from each probe. The basic process is as follows. A charge signal is generated when the sensor is exposed to medium-energy protons. The charge pulse is converted into a voltage signal through the pre-amplifier and the main amplifier. Meanwhile, the signal is pulse shaped. Subsequently, the pulse signal is collected by the AD (analog-to-digital) converter inside the data-processing unit. The pulse height represents the incident energy of the particle. After that, the FPGA (Field Programmable Gate Array) conducts amplitude analysis on the pulse height collected by AD to determine which energy level the incident particle belongs to.

*4.4. Medium-Energy Electron Detection Unit*

The basic detection principle of medium-energy electron detection is similar to that of medium-energy protons. Three probes are also used to measure electrons in nine directions. The differences between the MEEDU and the MEPDU include the geometric factor setting [44] of the instrument and the use of no deflection magnets. The MEEDU's specific structure is shown in Figure 9.

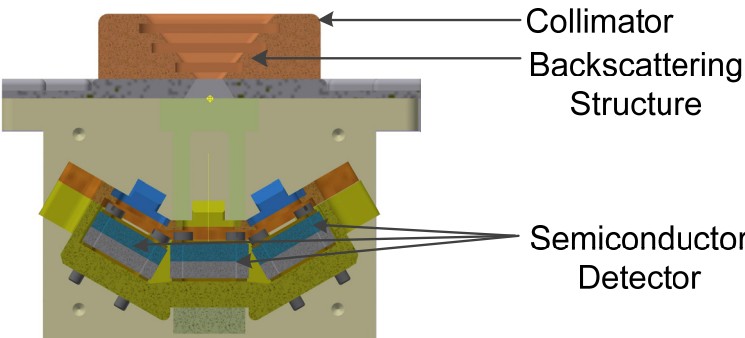

**Figure 9.** Probe section of MEEDU.

The geometric factor of the instrument limits the particle flux entering the detector. The flux of medium-energy protons and medium-energy electrons in the orbit of the space station are not the same. Hence, the geometric factor settings are also different. The deflection magnet is mainly used in the MEPDU to deflect electrons. In contrast, the magnet design is canceled in the medium-energy electron detection unit. Hence,

both medium-energy electrons and medium-energy protons can reach the detector. In the design, a detector with a thick light-blocking coating is selected for medium-energy electron detection. Due to the weak penetrative ability of medium-energy protons, low-energy end protons can be absorbed by the coating, thereby reducing the interference of medium-energy protons in medium-energy electron signals. Meanwhile, the medium-energy proton flux is much smaller than the medium-energy electron flux. Thus, the measurement error can be guaranteed to be better than 20%.

### 4.5. Shared Data Management Unit

The EPD is electrically connected to the space station platform. Specifically, the internal SDMU of the payload is responsible for power supply and distribution management as well as data transmission. The power distribution equipment on the Wentian laboratory cabin module provides working power for the payload. The EPD is interfaced with the application information system of the Wentian laboratory cabin module through the 1553B and FC-AE-1553 bus line, and it is responsible for completing the space environment command injection, data storage, and forwarding downlink.

The detection data are sent to the space station platform by being downloaded through the FC-AE-1553 bus line. To achieve the purpose of fast guarantee during data processing, the data packets are divided into quick-view data and level-I data. The quick-view data includes typical space particle environmental data that are harmful to the space station, such as dose rate and high-energy proton flux. Level-I data maintain the initial information of the particles to the greatest extent and can be analyzed in more detail on the ground after downloading. All the data are sent to the scientific personnel. After quick-view analysis and preliminary processing of the data, the working status, that is, the working mode and threshold of the instrument, can be adjusted. The schematic diagram of the EPD's electronics principle is shown in Figure 10.

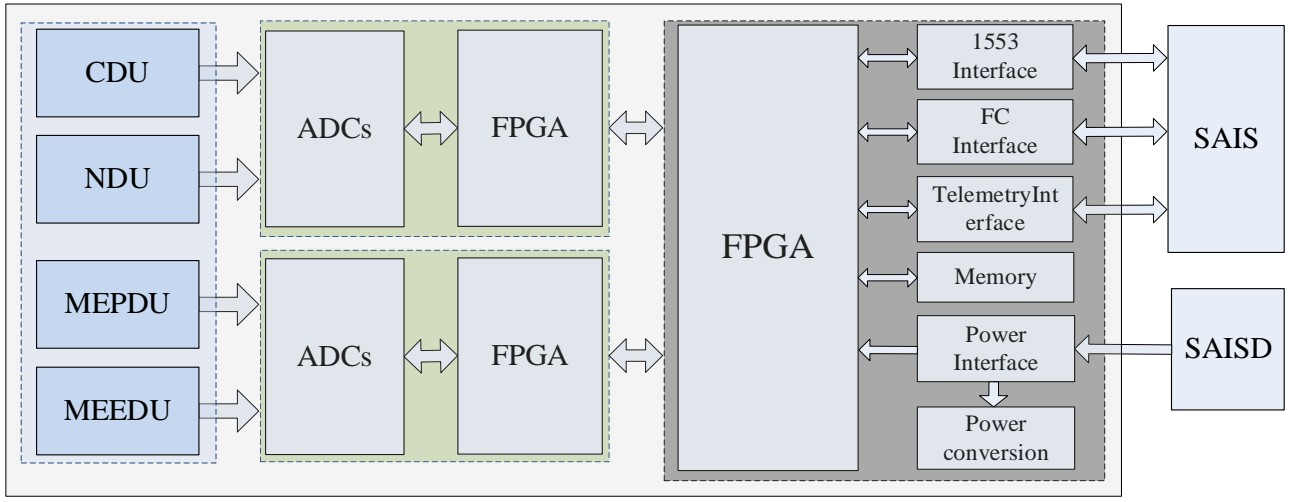

CDU = Comprehensive detection unit
NDU = Neutron detection unit
MEPDU = Medium-energy proton detection unit
MEEDU = Medium-energy electron detection unit

SAIS = Space application information system
SAISD = Space application information system distributor

**Figure 10.** Schematic diagram of the EPD's electronics principle.

## 5. Ground Calibration Verification

### 5.1. Calibration Contents

After completing the development of the formal product of the EPD on the Wentian laboratory cabin module, to verify and accurately present the actual detection indicators, ground calibration verification work is conducted on various technical indicators, including energy resolution, linearity, actual demarcation points of energy level, division precision

of energy level, measurement precision of flux, particle type identification ability, LET spectrum measurement range and measurement precision, and dose rate range.

- Calibration of energy resolution and measurement precision

Energy resolution is designed to measure the level of the performance parameters of the instrument itself, and to investigate the evaluation of the instrument's measurement repeatability of particles with a single energy. The higher the energy resolution of the instrument is, the greater the measurement precision can be.

- Energy range calibration

The purpose of energy range calibration is to confirm the actual measurement range of the instrument in each energy level through calibration. The design and debugging of the instrument are conducted according to the requirements of the user's indicators. However, after the actual development of the instrument, its indicators need to be confirmed to obtain the real energy measurement range of the instrument.

- Flux precision calibration

The purpose of flux calibration is to calibrate the actual counting ability of the particle detector, test the actual counting ability range of the detector and the flux test precision, obtain the corresponding precision of the detector to the flux (that is, the relationship between detector output and incident particle count), determine the detection sensitivity, and finally, evaluate the impact of the background noise count on the detector.

- Calibration of charged particle identification ability

The comprehensive measurement unit distinguishes charged particles including electrons, protons, and Z < 26 heavy ions. The calibration of charged particle identification ability can reflect the ability of the instrument to identify charged particle types and estimate the probability of mixing between different particles.

- Calibration of dose rate and LET spectrum range

Dose rate and LET spectrum are important indicators reflecting the radiation intensity in space and are of great significance to the study of the space radiation environment and radiation protection. Through the calibration test, we verify whether the designed dose rate range and LET spectrum range meet the indicator requirements.

- Calibration of neutron identification ability

Although the neutron detection unit can eliminate the interference of charged particles through the anticoincidence structure, it cannot distinguish neutrons and gamma radiation from neutral particles. The discrimination of neutrons and gamma radiation depends on their pulse shapes in CLYC. The calibration of the pulse shape discrimination (PSD) ability can reflect the ability of the instrument to identify neutrons.

### 5.2. Calibration Results

Since different detection units measure different physical quantities, different detection units adopt different calibration methods (Table 2). For the calibration of energy particle detectors, if the beam conditions were not satisfied, the method of combining equivalent signal source calibration and simulation analysis would be adopted.

Through the accelerator calibration experiments, the calibration of the equivalent signal source, and the analysis and processing of the simulation data of the EPD on the Wentian laboratory cabin module of the space station, the instrument energy resolution, linearity, actual demarcation point of energy level, division precision of energy level, measurement precision of flux, particle type identification ability, LET spectrum measurement range and measurement precision, and dose rate range are all realized. It should be noted that, during accelerator calibration, the output beam flux or beam geometry can be adjusted by various means. Generally, it is necessary to ensure that the particle count reaching the sensor is between tens and hundreds of thousands per second. In addition, the linear

extrapolation method is used for medium proton calibration with energy greater than 500 keV of MEPDU, because of the limited energy of particles accelerator available. Finally, the compliance of the obtained instrument indicators is shown in Table 3.

**Table 2.** Calibration modes of different detection units.

| Detection Unit | Calibration Condition |
|---|---|
| MEEDU | Beijing Huairou Electron Accelerator, Chinese Academy of Sciences<br>Beam range: 20 keV–1.6 MeV |
| MEPDU | For ≤500 keV: Beijing Huairou Electron Accelerator, Chinese Academy of Sciences<br>For >500 keV: linear extrapolation |
| CDU | (1) Linearity:<br>Electron and proton (<1.6 MeV): Beijing Huairou Electron Accelerator<br>Proton (15 to 20 MeV): Proton and Heavy Ion Accelerator (401) of China Atomic Energy Academy<br>For other gains: Equivalent signal calibration<br>(2) Energy measurement range, LET spectrum, and dose rate: Electron accelerator, Equivalent signals, and Simulation analysis<br>(3) Particle composition identification: Accelerator calibration combined with Simulation calculation |
| NDU | (1) Particle type identification: $^{252}$Cf<br>(2) Neutron energy range: China Dongguan hash source (neutron)<br>(3) Linearity calibration: $^{137}$Cs/$^{60}$Co/$^{252}$Cf |

**Table 3.** Compliance of indicator calibration test.

| Technical Indicators | Actual Product Indicators | Test Verification |
|---|---|---|
| Protons:<br>20 keV–300 MeV;<br>Nine directions for medium energy;<br>Five directions for high energy | 19.58 keV–300.3 MeV;<br>Nine directions for medium energy;<br>Five directions for high energy | Combination of Beijing Huairou accelerator, proton and heavy ion (401) accelerator, and equivalent signal calibration |
| Heavy ions: 8 MeV/n–400 MeV/n;<br>Five directions | 7.5 MeV/n–401.7 MeV/n;<br>Five directions | Combination of Beijing Huairou accelerator, 401 accelerator, and equivalent signal calibration |
| Electrons:<br>20 keV–10 MeV;<br>Five directions for medium energy;<br>Five directions for high energy | 19.47 keV–15 MeV;<br>Nine directions of medium energy;<br>Five directions of high energy | Combination of Beijing Huairou accelerator, 401 accelerator, and equivalent signal calibration |
| Neutrons:<br>0.025 eV–100 MeV | 0.025 eV–110 MeV | China Dongguan hash source (<110 MeV) |
| LET spectral range:<br>0.233~17,475 keV/μm;<br>>17,475 keV/μm | 0.233~17,575 keV/μm;<br>>17,575 keV/μm | Beijing Huairou accelerator test, 401 accelerator combined with equivalent signal test |
| Dose rate: 0.1~1000 mGy/day | 0.09~1000.98 mGy/day | Beijing Huairou accelerator test, 401 accelerator combined with equivalent signal test |

As examples, the calibration test results of heavy ion species identification and neutron/gamma identification of EPD are briefly listed below. Figure 11 shows the calibration result of charged particles identification ability. We used electron, proton, and Li for this calibration with Beijing Huairou Electron Accelerator and Proton and Heavy Ion Accelerator (401) of the China Atomic Energy Academy. Based on the measured results of the experiment, we analyzed the deposition energy of different types of particles in each sensor of the instrument.

In the above figure, B represents the energy loss in the second detector, while T means the energy loss in all instruments. Due to the limited types of accelerator particles available, the GEANT4 Monte Carlo simulation method is used in the extrapolation process for other particle types. This report recommends establishing a simulation model of the CDU in GEANT4 to simulate different types of particle responses and using limited measured data

to correct the simulation results. According to calculations, the probability of iron being mistaken for oxygen is 1.33%.

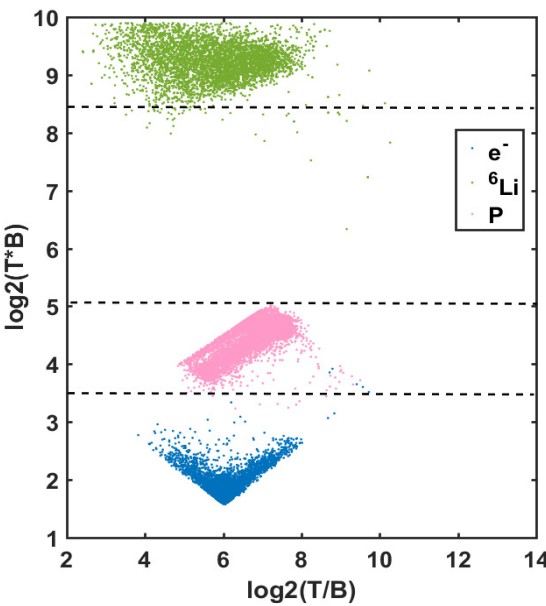

**Figure 11.** The calibration result of charged particle identification ability.

Figure 12 bellow shows the calibration result of neutron/gamma identification which the pulse shape data obtained from the irradiation of the $^{252}$Cf neutron source.

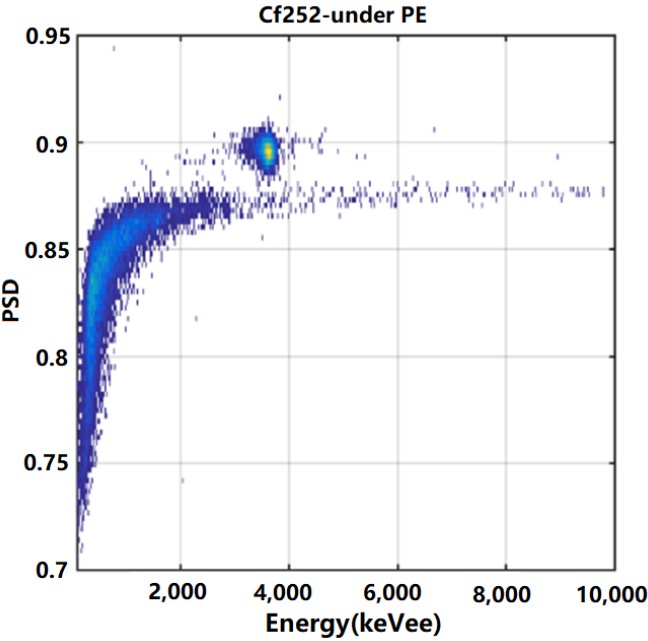

**Figure 12.** The calibration result of neutron/gamma identification.

## 6. Conclusions

The EPD on the Wentian laboratory cabin module of the Tiangong space station went through three stages during the ground development processes: planning, initial prototype, and formal prototype. It incorporates key technologies, including high-precision particle type identification, n/$\gamma$ discrimination, and wide-energy spectrum measurement range. It also passes a series of ground performance tests. For the first time, we realized the

comprehensive detection of the particle radiation environment, that is, the types of particle radiation in space station orbit including electrons, protons, heavy ions, and neutrons, the measurement range from 20 keV for medium energy electron proton to heavy ion GeV, and the radiation LET spectrum and dose rate.

On 24 July 2022, the EPD was successfully launched into space with the Wentian laboratory cabin module. As China's first space station standard payload, the EPD was grabbed by a mechanical arm from the cargo airlock cabin and successfully installed on the exposed platform outside the cabin on 7 January 2023. Since then, it has started the continuous detection of various charged particles, neutrons, and particle radiation effects in the orbit of the space station, which will last for a period of ten years, providing key data support for the space station, astronauts, and related research on space science.

**Author Contributions:** Conceptualization, G.S. and S.Z.; methodology, G.S. and S.Z.; software, B.Y.; validation, D.H. and Z.Q.; formal analysis, D.H.; investigation, S.Z. and D.H.; resources, H.Z.; data curation, X.Z. and C.W.; writing—original draft preparation G.S. and Z.Y.; writing—review and editing, G.S.; supervision, Y.S.; project administration, Y.S.; funding acquisition, X.Z. All authors have read and agreed to the published version of the manuscript.

**Funding:** This research received no external funding.

**Institutional Review Board Statement:** Not applicable.

**Informed Consent Statement:** Not applicable.

**Data Availability Statement:** No new data were created or analyzed in this study. Data sharing is not applicable to this article.

**Conflicts of Interest:** The authors declare no conflict of interest.

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
