# Peer review of "Using Energy Particle Detection Technology on the Tiangong’s Space Station’s Wentian Laboratory Cabin Module"

_aerospace, doi:10.3390/aerospace10040373_

Round 1

Reviewer 1 Report

The proposed paper is interesting and worth publication but major revision should be done before publication:

--The type of particles and energy range expected to be detected are repeated several time in the paper.

--Chapter 2 is probably too long. A lot of the given information can be found in textbooks and are not novelties. Therefore authors should focus on novelties and give more details on the work/measurements done during the research

-- Comparison should be done with existing instrument of the same type (i.e. ESA HMRM)

-- In chapter 4.1, on figure 5 it is not clear where parts A, B, C & D are located.

·--The parameters of equation (1) line 293 are not all defined: v , I

--The equation (1) line 293 is too generic and depend on type of particle and interaction type (i.e. NIEL)

--Figure 6 horizontal axis title must be translated.

--On line 319 authors write that dose measurement is done for silicon, which consistent with the device but authors also mention several times that effects on astronauts are important. At least some details should be given for dose conversion from silicon to water

·--On line 351: is this count rate ?

--On line 361: “…high speed…” authors should give values; it is important to know the capabilities of the instrument. I.e. can the instrument be used to studied SEP ?

-- More numerical data should be given on the rejection rate capabilities Vs type and energies of particles for all the sub units (MEPDU, MEEDU, NDU…)

--More numerical data should be given on the experimental condition (flux used during measurement, beam geometry …)

-- On Figure 12, T/B should be defined for the horizontal axis title

Author Response

Your comments have been very helpful in improving the paper, and I have completed and revised it as required. You can see the attachment for modifications.

Thank you very much.

Reviewer 2 Report

This paper documents the design and development of a new space radiation detector deployed in low Earth orbit. This detector is a major addition to the suite of radiation instruments operating in LEO, and the description in this paper will be of great value to future researchers. I recommend publication after a number of revisions listed below.

l. 44 ff. – recommend using the more common SI units of Gy (or mGy) for absorbed dose and keV/micron for LET.

l. 49 – it was not clear on first reading (until the Conclusions) that the detector is operating on the outside of the module. It should be stated at or near the introduction.

Fig. 1 – It would be useful to have a photo of the instrument as deployed on the module.

l. 58 – define “quiet time” (which I assume refers to solar activity)

l. 67 – “…11 year solar cycle…”

l. 81-82 and 107 ff. --  the text should make a clear distinction between the relative contributions of SAA protons and GCR to particle flux, absorbed dose and dose equivalent

l. 95 – delete “in galaxies”. Cosmic rays originate both within our galaxy (“GCR”) and extragalactically (“EGCR”).

l. 97 – change “solar magnetic field” to “interplanetary magnetic field and the solar wind”

l. 99 – change “still unclear” to “not fully understood”.

Fig. 2 should be cited as from:  J.A. Simpson Elemental and isotopic composition of the galactic cosmic rays Ann. Rev. Nucl. Part Sci., 33 (1983), pp. 323-382.

l. 117 – change “high solar years than in low solar years” to  “periods of high solar activity”.

l. 152-157 – This paragraph is rather awkward. I suggest replacing it by something like: “In addition, space radiation is known or hypothesized  to cause deleterious effects to the central nervous and cardiovascular systems, as well as induction of late cancers” and replacing Refs. 25-27 with one of the many comprehensive references such as NCRP Report 132 or  G.A. Nelson, Radiation Research, 185(4):349-358 (2016).

Fig. 5 – Labels A, B, C, E1, E2 are missing

l. 253 ff. – What are the thicknesses and areal dimensions of the various detectors?

Fig. 6 – is not referenced in the text.

l. 468 ff – What is the resolution in Z?

Table 2. – What is the justification for the linear extrapolation for > 500 keV? Also, while I realize that it would add considerable length to the paper, additional detail on the calibrations would be useful.

l. 499 ff. – Given the importance of GCR exposures, more detail should be provided on the heavy ion calibrations. How many different ion-energy combinations were used in calibrating, and how were the results interpolated/extrapolated? How were the data in Fig. 12 obtained,,and what do the axis labels denote? What is the charge resolution between O and Fe?

Author Response

(The authors gave the same response as above.)

Round 2

Reviewer 1 Report

At the end of chapter 4.2, you write : "...The output is sent o the data processing circuit for high-speed sampling with the frequensy of 500 MHz per second...".

I do not understand the meaning a frequency in Hz/second, maybe it is only MHz ? 

Author Response

Thank you very much.

Reviewer 2 Report

Thank you for addressing the suggested revisions.

Author Response

Thank you very much.